# Zero-shot Synthesis with Group-Supervised Learning

**Yunhao Ge, Sami Abu-El-Haija, Gan Xin, Laurent Itti**
University of Southern California
`yunhaoge@usc.edu, sami@haija.org, gxin@usc.edu, itti@usc.edu`

## Abstract

Visual cognition of primates is superior to that of artificial neural networks in its ability to "envision" a visual object, even a newly-introduced one, in different *attributes* including pose, position, color, texture, etc. To aid neural networks to envision objects with different attributes, we propose a family of objective functions, expressed on *groups of examples*, as a novel learning framework that we term Group-Supervised Learning (GSL). GSL allows us to decompose inputs into a disentangled representation with swappable components, that can be recombined to synthesize new samples. For instance, images of *red boats* & *blue cars* can be decomposed and recombined to synthesize novel images of *red cars*. We propose an implementation based on auto-encoder, termed group-supervised zero-shot synthesis network (GZS-Net) trained with our learning framework, that can produce a high-quality *red car* even if no such example is witnessed during training. We test our model and learning framework on existing benchmarks, in addition to a new dataset that we open-source. We qualitatively and quantitatively demonstrate that GZS-Net trained with GSL outperforms state-of-the-art methods.

## 1 Introduction

Primates perform well at generalization tasks. If presented with a single visual instance of an object, they often immediately can generalize and envision the object in different attributes, e.g., in different 3D pose (Logothetis et al., 1995). Primates can readily do so, as their previous knowledge allows them to be cognizant of attributes. Machines, by contrast, are most-commonly trained on sample features (e.g., pixels), not taking into consideration attributes that gave rise to those features.

To aid machine cognition of visual object attributes, a class of algorithms focuses on *learning disentangled representations* (Kingma & Welling, 2014; Higgins et al., 2017; Burgess et al., 2018; Kim & Mnih, 2018; Chen et al., 2018), which map visual samples onto a latent space that separates the information belonging to different attributes. These methods show disentanglement by interpolating between attribute values (e.g., interpolate pose, etc). However, these methods usually process one sample at a time, rather than contrasting or reasoning about a group of samples. We posit that semantic links across samples could lead to better learning.

We are motivated by the visual generalization of primates. We seek a method that can synthesize realistic images for arbitrary queries (e.g., a particular car, in a given pose, on a given background), which we refer to as *controlled synthesis*. We design a method that enforces semantic consistency of attributes, facilitating controlled synthesis by leveraging semantic links between samples. Our method maps samples onto a disentangled latent representation space that (i) consists of subspaces, each encoding one attribute (e.g., identity, pose, ...), and, (ii) is such that two visual samples that share an attribute value (e.g., both have identity "car") have identical latent values in the shared attribute subspace (identity), even if other attribute values (e.g., pose) differ. To achieve this, we propose a general learning framework: *Group Supervised Learning* (GSL, Sec. 3), which provides a learner (e.g., neural network) with groups of semantically-related training examples, represented as *multigraph*. Given a query of attributes, GSL proposes groups of training examples with attribute combinations that are useful for synthesizing a test example satisfying the query (Fig. 1). This endows the network with an envisioning capability. In addition to applications in graphics, controlled synthesis can also augment training sets for better generalization on machine learning tasks (Sec. 6.3).

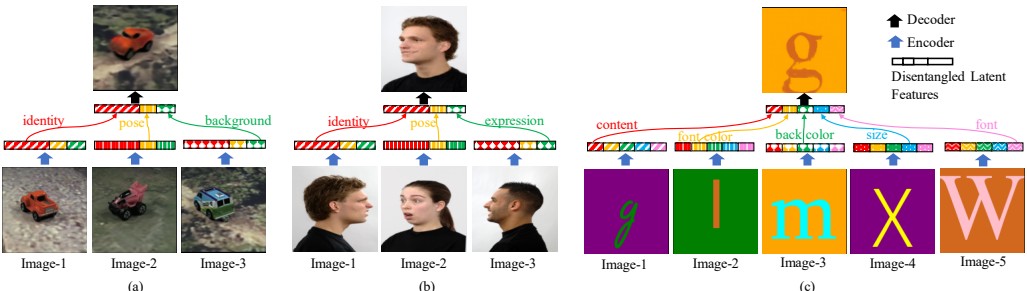

Figure 1: Zero-shot synthesis performance of our method. (a), (b), and (c) are from datasets, respectively, iLab-20M, RaFD, and Fonts. Bottom: training images (attributes are known). Top: Test image (attributes are a *query*). Training images go through an encoder, their latent features get combined, passed into a decoder, to synthesize the requested image. Sec. 4.2 shows how we disentangle the latent space, with explicit latent feature swap during training.

As an instantiation of GSL, we propose an encoder-decoder network for zero-shot synthesis: *Group-Supervised Zero-Shot Synthesis Network* (GZS-Net, Sec. 4). While learning (Sec. 4.2 & 4.3), we repeatedly draw a group of semantically-related examples, as informed by a multigraph created by GSL. GZS-Net encodes group examples, to obtain latent vectors, then *swaps* entries for one or more attributes in the latent space *across examples*, through multigraph edges, then decodes into an example within the group (Sec. 4.2).

Our contributions are: (i) We propose Group-Supervised Learning (GSL), explain how it casts its admissible datasets into a multigraph, and show how it can be used to express learning from semantically-related groups and to synthesize samples with controllable attributes; (ii) We show one instantiation of GSL: Group-supervised Zero-shot Synthesis Network (GZS-Net), trained on groups of examples and reconstruction objectives; (iii) We demonstrate that GZS-Net trained with GSL outperforms state-of-the-art alternatives for controllable image synthesis on existing datasets; (iv) We provide a new dataset, Fonts[1], with its generating code. It contains 1.56 million images and their attributes. Its simplicity allows rapid idea prototyping for learning disentangled representations.

## 2 RELATED WORK

We review research areas, that share similarities with our work, to position our contribution.

**Self-Supervised Learning** (e.g., Gidaris et al. (2018)) admits a dataset containing features of training samples (e.g., upright images) and maps it onto an auxiliary task (e.g., rotated images): dataset examples are drawn and a random transformation (e.g., rotate 90°) is applied to each. The task could be to predict the transformation (e.g., =90°) from the transformed features (e.g., rotated image). Our approach is similar, in that it also creates auxiliary tasks, however, the tasks we create involve semantically-related *group of examples*, rather than from one example at a time.

**Disentangled Representation Learning** are methods that infer *latent factors* given example visible features, under a generative assumption that each latent factor is responsible for generating one semantic attribute (e.g. color). Following Variational Autoencoders (VAEs, Kingma & Welling, 2014), a class of models (including, Higgins et al., 2017; Chen et al., 2018) achieve disentanglement *implicitly*, by incorporating into the objective, a distance measure e.g. KL-divergence, encouraging the latent factors to be statistically-independent. While these methods can disentangle the factors *without* knowing them beforehand, unfortunately, they are unable to generate novel combinations not witnessed during training (e.g., generating images of *red car*, without any in training). On the other hand, our method *requires* knowing the semantic relationships between samples (e.g., which objects are of same identity and/or color), but can then synthesize novel combinations (e.g., by stitching latent features of "any car" plus "any red object").

**Conditional synthesis** methods can synthesize a sample (e.g., image) and some use information external to the synthesized modalities, e.g., natural language sentence Zhang et al. (2017); Hong et al.

---

[1]http://ilab.usc.edu/datasets/fonts

(2018) or class label Mirza & Osindero (2014); Tran et al. (2017). Ours differ, in that our "external information" takes the form of semantic relationships between samples. There are methods based on GAN Goodfellow et al. (2014) that also utilize semantic relationships including Motion Re-targeting (Yang et al., 2020), which unfortunately requires domain-specific hand-engineering (detect and track human body parts). On the other hand, we design and apply our method on different tasks (including people faces, vehicles, fonts; see Fig. 1). Further, we compare against two recent GAN methods starGAN (Choi et al., 2018) and ELEGANT (Xiao et al., 2018), as they are state-of-the-art GAN methods for amending visual attributes onto images. While they are powerful in carrying local image transformations (within a small patch, e.g., changing skin tone or hair texture). However, our method better maintains global information: when rotating the main object, the scene also rotates with it, in a semantically coherent manner. Importantly, our learning framework allows expressing simpler model network architectures, such as feed-forward auto-encoders, trained with only reconstruction objectives, as opposed to GANs, with potential difficulties such as lack of convergence guarantees.

**Zero-shot learning** also consumes side-information. For instance, models of Lampert (2009); Atzmon & Chechik (2018) learn from object attributes, like our method. However, (i) these models are supervised to accurately predict attributes, (ii) they train and infer one example at a time, and (iii) they are concerned with classifying unseen objects. We differ in that (i) no learning gradients (supervision signal) are derived from the attributes, as (ii) these attributes are used to group the examples (based on shared attribute values), and (iii) we are concerned with generation rather than classification: we want to synthesize an object in previously-unseen attribute combinations.

**Graph Neural Networks** (GNNs) (Scarselli et al., 2009) are a class of models described on graph structured data. This is applicable to our method, as we propose to create a multigraph connecting training samples. In fact, our method can be described as a GNN, with *message passing* functions (Gilmer et al., 2017) that are aware of the latent space partitioning per attribute (explained in Sec. 4). Nonetheless, for self-containment, we introduce our method in the absence of the GNN framework.

## 3 GROUP-SUPERVISED LEARNING

### 3.1 DATASETS ADMISSIBLE BY GSL

Formally, a dataset admissible by GSL containing $n$ samples $\mathcal{D} = \{x^{(i)}\}_{i=1}^{n}$ where each example is accompanied with $m$ attributes $\mathcal{D}_a = \{(a_1^{(i)}, a_2^{(i)}, \ldots a_m^{(i)})\}_{i=1}^{n}$. Each attribute value is a member of a countable set: $a_j \in \mathcal{A}_j$. For instance, pertaining to visual scenes, $\mathcal{A}_1$ can denote foreground-colors $\mathcal{A}_1 = \{\text{red}, \text{yellow}, \ldots\}$, $\mathcal{A}_2$ could denote background colors, $\mathcal{A}_3$ could correspond to foreground identity, $\mathcal{A}_4$ to (quantized) orientation. Such datasets have appeared in literature, e.g. in Borji et al. (2016); Matthey et al. (2017); Langner et al. (2010); Lai et al. (2011).

### 3.2 AUXILIARY TASKS VIA MULTIGRAPHS

Given a dataset of $n$ samples and their attributes, we define a multigraph $M$ with node set $[1..n]$. Two nodes, $i, k \in [1..n]$ with $i \neq k$ are connected with edge labels $M(i, k) \subseteq [1..m]$ as:

$$M(i, k) = \left\{ j \mid a_j^{(i)} = a_j^{(k)}; j \in [1..m] \right\}.$$

In particular, $M$ defines a multigraph, with $|M(i, k)|$ denoting the number of edges connecting nodes $i$ and $k$, which is equals the number of their shared attributes. Fig. 2 depicts a (sub-)multigraph for the Fonts dataset (Sec. 5.1).

**Definition 1** COVER($S, i$): *Given node set $S \subseteq [1..|\mathcal{D}_g|]$ and node $i \in [1..|\mathcal{D}_g|]$ we say set $S$ covers node $i$ if every attribute value of $i$ is in at least one member of $S$. Formally:*

$$\text{COVER}(S, i) \iff [1..m] = \bigcup_{k \in S} M(i, k). \tag{1}$$

When COVER($S, i$) holds, there are two mutually-exclusive cases: either $i \in S$, or $i \notin S$, respectively shaded as green and blue in Fig. 2 (b). The first case trivially holds even for small $S$, e.g. COVER($\{i\}, i$) holds for all $i$. However, we are interested in non-trivial sets where $|S| > 1$, as sets with $|S| = 1$ would cast our proposed network (Sec. 4) to a standard Auto-Encoder. The second case

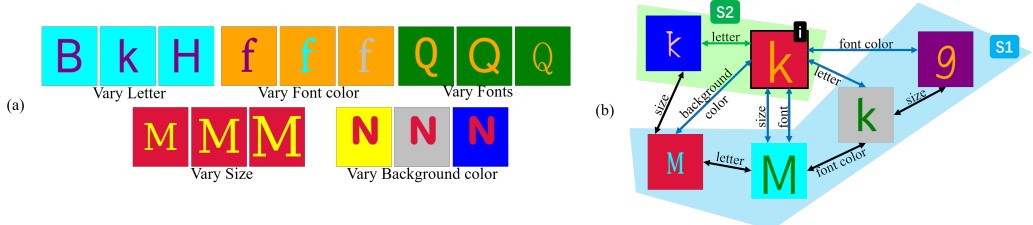

Figure 2: (a) Samples from our proposed Fonts dataset, shown in groups. In each group, we vary one attribute but keep others the same. (b) (Sub-)multigraph of our Fonts dataset. Each edge connect two examples sharing an attribute. Sets $S_1$ and $S_2$ *cover* sample $i$.

is crucial for zero-shot synthesis. Suppose the (image) features of node $i$ (in Fig. 2 (b)) are not given, we can search for $S_1$, under the assumption that if $\text{COVER}(S_1, i)$ holds, then $S_1$ contains sufficient information to synthesize $i$'s features as they are not given ($i \notin S_1$).

Until this point, we made no assumptions how the pairs $(S, i)$ are extracted (mined) from the multigraph s.t. $\text{COVER}(S, i)$ holds. In the sequel, we train with $|S| = 2$ and $i \in S$. We find that this particular specialization of GSL is easy to program, and we leave-out analyzing the impact of mining different kinds of cover sets for future work.

## 4 GROUP-SUPERVISED ZERO-SHOT SYNTHESIS NETWORK (GZS-NET)

We now describe our ingredients towards our goal: synthesize holistically-semantic novel images.

### 4.1 AUTO-ENCODING ALONG RELATIONS IN $M$

Auto-encoders $(D \circ E) : \mathcal{X} \to \mathcal{X}$ are composed of an encoder network $E : \mathcal{X} \to \mathbb{R}^d$ and a decoder network $D : \mathbb{R}^d \to \mathcal{X}$. Our networks further utilize $M$ emitted by GSL. GZS-Net consists of

$$\text{an encoder} \quad E : \mathcal{X} \times \mathcal{M} \to \mathbb{R}^d \times \mathcal{M} \quad ; \quad \text{and a decoder} \quad D : \mathbb{R}^d \times \mathcal{M} \to \mathcal{X}. \tag{2}$$

$\mathcal{M}$ denotes the space of sample pairwise-relationships. GSL realizes such $(X, M) \subset \mathcal{X} \times \mathcal{M}$, where $X$ contains (a batch of) training samples and $M$ the (sub)graph of their pairwise relations. Rather than passing as-is the output of $E$ into $D$, one can modify it using algorithm $A$ by chaining: $D \circ A \circ E$. For notation brevity, we fold $A$ into the encoder $E$, by designing a **swap** operation, next.

### 4.2 DISENTANGLEMENT BY **SWAP** OPERATION

While training our auto-encoder $D(E(X, M))$, we wish to disentangle the latents output by $E$, to provide use for using $D$ to decode samples not given to $E$. $D$ (/ $E$) outputs (/ inputs) **one or more** images, onto (/ from) the image space. Both networks can access feature and relationship information.

At a high level, GZS-Net aims to *swap attributes* across images by swapping corresponding entries across their latent representations. Before any training, we fix partitioning of the the latent space $Z = E(X, M)$. Let row-vector $z^{(1)} = [g_1^{(1)}, g_2^{(1)}, \ldots, g_m^{(1)}]$ be the concatenation of $m$ row vectors $\{g_j^{(1)} \in \mathbb{R}^{d_j}\}_{j=1}^m$ where $d = \sum_{j=1}^m d_j$ and the values of $\{d_j\}_{j=1}^m$ are hyperparameters.

To simplify the notation to follow, we define an operation **swap** : $\mathbb{R}^d \times \mathbb{R}^d \times [1..m] \to \mathbb{R}^d \times \mathbb{R}^d$, which accepts two latent vectors (e.g., $z^{(1)}$ and $z^{(2)}$) and an attribute (e.g., 2) and returns the input vectors except that the latent features corresponding to the attribute are swapped. E.g.,

$$\mathbf{swap}(z^{(1)}, z^{(2)}, 2) = \mathbf{swap}([g_1^{(1)}, g_2^{(1)}, g_3^{(1)}, \ldots, g_m^{(1)}], [g_1^{(2)}, g_2^{(2)}, g_3^{(2)}, \ldots, g_m^{(2)}], 2)$$
$$= [g_1^{(1)}, g_2^{(2)}, g_3^{(1)}, \ldots, g_m^{(1)}], [g_1^{(2)}, g_2^{(1)}, g_3^{(2)}, \ldots, g_m^{(2)}]$$

**One-Overlap Attribute Swap**. To encourage disentanglement in the latent representation of attributes, we consider group $S$ and example $x$ s.t. $\text{COVER}(S, x)$ holds, and for all $x^o \in S, x \neq x^o$, the

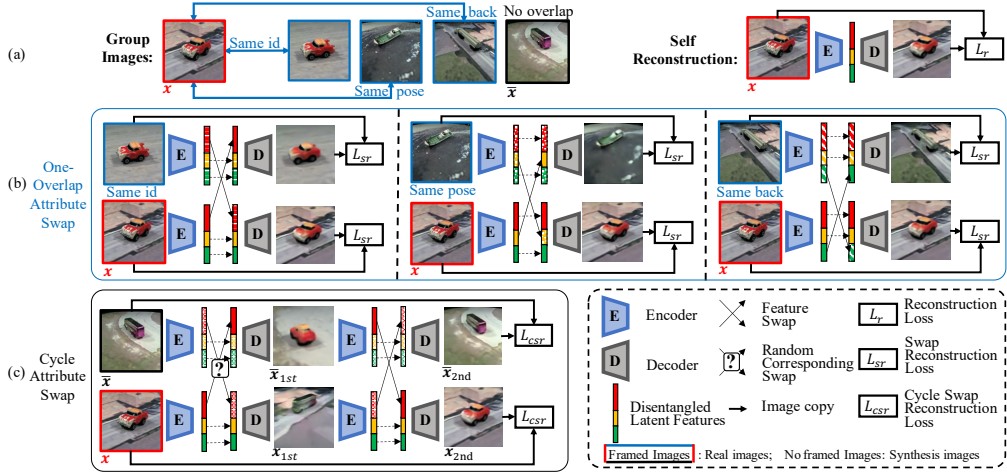

Figure 3: Architecture of GZS-Net, consisting of an encoder $E$: maps sample onto latent vector, and a decoder $D$: maps latent vector onto sample. The latent space is pre-partitioned among the attribute classes (3 shown: *identity*, *pose*, *background*). (a, left) considered examples: a center image ($x$, red border) and 3 images sharing one attribute with it, as well as a *no overlap* image sharing no attributes ($\bar{x}$, black border). (a, right) standard reconstruction loss, applied for all images. (b) **One-overlap attribute swap:** Two images with identical values for one attribute should be reconstructed into nearly the original images when the latent representations for that attribute are swapped ("no-op" swap; left: identity; middle: pose; right: background). (c) **Cycle swap**: given any example pair, we randomly pick an attribute class $j$. We encode both images, swap representations of $j$, decode, re-encode, swap on $j$ again (to reverse the first swap), and decode to recover the inputs. This unsupervised cycle enforces that double-swap on $j$ does not destroy information for other attributes.

pair $(x^o, x)$ share **exactly one** attribute value ($|M(x^o, x)| = 1$). Encoding those pairs, swapping the latent representation of the attribute, and decoding should then be a no-op if the swap did not affect other attributes (Fig. 3b). Specifically, we would like for a pair of examples, $x$ (red border in Fig. 3b) and $x^o$ (blue border) sharing only attribute $j$ (e.g., identity)[2], with $z = E(x)$ and $z^o = E(x^o)$, be s.t.

$$D\left(z_s\right) \approx x \quad \text{and} \quad D\left(z_s^{(o)}\right) \approx x^{(o)}; \quad \text{with} \ z_s, z_s^{(o)} = \textbf{swap}(z, z^o, j). \tag{3}$$

If, for each attribute, sufficient sample pairs share only that attribute, and Eq. 3 holds for all with zero residual loss, then disentanglement is achieved for that attribute (on the training set).

**Cycle Attribute Swap**. This operates on all example pairs, regardless of whether they share an attribute or not. Given two examples and their corresponding latent vectors, if we swap latent information corresponding to *any attribute*, we should end up with a sensible decoding. However, we may not have ground-truth supervision samples for swapping all attributes of all pairs. For instance, when swapping the *color attribute* between pair *orange truck* and *white airplane*, we would like to learn from this pair, even without any *orange airplanes* in the dataset. To train from any pair, we are motivated to follow a recipe similar to CycleGAN (Zhu et al., 2017). As shown in Fig. 3c, given two examples $x$ and $\bar{x}$: (i) sample an attribute $j \sim \mathcal{U}[1..m]$; (ii) encode both examples, $z = E(x)$ and $\bar{z} = E(\bar{x})$; (iii) swap features corresponding to attribute $j$ with $z_s, \bar{z}_s = \textbf{swap}(z, \bar{z}, j)$; (iv) decode, $\widehat{x} = D(z_s)$ and $\widehat{\bar{x}} = D(\bar{z}_s)$; (v) on a second round (hence, cycle), encode again as $\widehat{z} = E(\widehat{x})$ and $\widehat{\bar{z}} = E(\widehat{\bar{x}})$; (vi) another swap, which should reverse the first swap, $\widehat{z}_s, \widehat{\bar{z}}_s = \textbf{swap}(\widehat{z}, \widehat{\bar{z}}, j)$; (vii) finally, one last decoding which should approximately recover the original input pair, such that:

$$D\left(\widehat{z}_s\right) \approx x \quad \text{and} \quad D\left(\widehat{\bar{z}}_s\right) \approx \bar{x}; \tag{4}$$

If, after the two encode-swap-decode, we are able to recover the input images, regardless of which attribute we sample, this implies that swapping one attribute does not destroy latent information for other attributes. As shown in Sec. 5, this can be seen as a data augmentation, growing the effective training set size by adding all possible new attribute combinations not already in the training set.

---

[2]It holds that COVER($\{x, x^o\}, x$) and COVER($\{x, x^o\}, x^o$)

---

**Algorithm 1:** Training Regime; for sampling data and calculating loss terms

---

**Input:** Dataset $\mathcal{D}$ and Multigraph $M$

**Output:** $L_\text{r}, L_\text{sr}, L_\text{csr}$

1   Sample $x \in \mathcal{D}, S \subset \mathcal{D}$ such that $\text{COVER}(S, x)$ and $|S| = m$ and $\forall k \in S, |M(x, k)| = 1$

2   **for** $x^{(o)} \in S$ **do**

3      $z \leftarrow E(x); \; z^{(o)} \leftarrow E(x^{(o)}); \; \left( z_s, z_s^{(o)} \right) \leftarrow \textbf{swap}(z, z^{(o)}, j)$

4      $L_\text{sr} \leftarrow L_\text{sr} + ||D\left(z_s\right) - x||_{l_1} + \left|\left|D\left(z_s^{(o)}\right) - x^{(o)}\right|\right|_{l_1}$    # Swap reconstruction loss

5   $\bar{x} \sim \mathcal{D}$ and $j \sim \mathcal{U}[1..m]$    # Sample for Cycle swap

6   $z \leftarrow E(x); \; \bar{z} \leftarrow E(\bar{x}); \; (z_s, \bar{z}_s) \leftarrow \textbf{swap}(z, \bar{z}, j); \; \widehat{x} \leftarrow D(z_s); \; \widehat{\bar{x}} \leftarrow D(\bar{z}_s)$

7   $\widehat{z} \leftarrow E(\widehat{x}); \; \widehat{\bar{z}} \leftarrow E(\widehat{\bar{x}}); \; (\widehat{z}_s, \widehat{\bar{z}}_s) \leftarrow \textbf{swap}(\widehat{z}, \widehat{\bar{z}}, j)$

8   $L_\text{csr} \leftarrow ||D\left(\widehat{z}_s\right) - x||_{l_1} + \left|\left|D\left(\widehat{\bar{z}}_s\right) - \bar{x}\right|\right|_{l_1}$    # Cycle reconstruction loss

9   $L_\text{r} \leftarrow ||D\left(E(x)\right) - x||_{l_1}$    # Standard reconstruction loss

---

## 4.3   Training and Optimization

Algorithm 1 lists our sampling strategy and calculates loss terms, which we combine into a total loss

$$\mathcal{L}(E, D; \mathcal{D}, M) = L_\text{r} + \lambda_\text{sr} L_\text{sr} + \lambda_\text{csr} L_\text{csr}, \tag{5}$$

where $L_\text{r}$, $L_\text{sr}$ and $L_\text{csr}$, respectively are the reconstruction, swap-reconstruction, and cycle construction losses. Scalar coefficients $\lambda_\text{sr}, \lambda_\text{csr} > 0$ control the relative importance of the loss terms. The total loss $\mathcal{L}$ can be minimized w.r.t. parameters of encoder ($E$) and decoder ($D$) via gradient descent.

## 5   Qualitative Experiments

We qualitatively evaluate our method on zero-shot synthesis tasks, and on its ability to learn disentangled representations, on existing datasets (Sec. 5.2), and on a dataset we contribute (Sec. 5.1).

**GZS-Net architecture.** For all experiments, the encoder $E$ is composed of two convolutional layers with stride 2, followed by 3 residual blocks, followed by a convolutional layer with stride 2, followed by reshaping the response map to a vector, and finally two fully-connected layers to output 100-dim vector as latent feature. The decoder $D$ mirrors the encoder, and is composed of two fully-connected layers, followed by reshape into cuboid, followed by de-conv layer with stride 2, followed by 3 residual blocks, then finally two de-conv layers with stride 2, to output a synthesized image.

### 5.1   Fonts Dataset & Zero-shot synthesis Performance

**Design Choices.** *Fonts* is a computer-generated image datasets. Each image is of an alphabet letter and is accompanied with its generating attributes: Letters (52 choices, of lower- and upper-case English alphabet); size (3 choices); font colors (10); background colors (10); fonts (100); giving a total of 1.56 million images, each with size $(128 \times 128)$ pixels. We propose this dataset to allow fast testing and idea iteration on zero-shot synthesis and disentangled representation learning. Samples from the dataset are shown in Fig. 2. Details and source code are in the Appendix. We partition the 100-d latents equally among the 5 attributes. We use a train:test split of 75:25.

**Baselines.** We train four baselines:

- The first three are a standard Autoencoder, a $\beta$-VAE (Higgins et al., 2017), and $\beta$-TCVAE (Chen et al., 2018). $\beta$-VAE and $\beta$-TCVAE show reasonable disentanglement on the dSprites dataset (Matthey et al., 2017). Yet, they do not make explicit the assignment between latent variables and attributes, which would have been useful for precisely controlling the attributes (e.g. color, orientation) of synthesized images. Therefore, for these methods, we designed a best-effort approach by *exhaustively* searching for the assignments. Once assignments are known, swapping attributes between images might become possible with these VAEs, and hopefully enabling for controllable-synthesis. We denote these three baselines with this **E**xhaustive **S**earch, using suffix **+ES**. Details on Exhaustive Search are in the Appendix.

Figure 4: Zero-shot synthesis performance compare on Fonts. 7-11 and 18-22 columns are input group images and we want to combine the specific attribute of them to synthesize an new images. 1-5 and 12-16 columns are synthesized images use auto-encoder + Exhaustive Swap (AE+ES), $\beta$-VAE + Exhaustive Swap ($\beta$-VAE+ES), $\beta$-TCVAE + Exhaustive Swap ($\beta$-TCVAE+ES), auto-encoder + Directly Supervision (AE+DS) and GZS-Net respectively. 6 and 17 columns are ground truth (GT)

- The fourth baseline, AE+DS, is an auto-encoder where its latent space is partitioned and each partition receives direct supervision from one attribute. Further details are in the Appendix.

As shown in Fig. 4, our method outperforms baselines, with second-runner being AE+DS: With discriminative supervision, the model focus on the most discriminative information, e.g., can distinguish e.g. across size, identity, etc, but can hardly synthesize photo-realistic letters.

## 5.2 ZERO-SHOT SYNTHESIS ON ILAB-20M AND RAFD

**iLab-20M** (Borji et al., 2016): is an attributed dataset containing images of toy vehicles placed on a turntable using 11 cameras at different viewing points. There are 3 attribute classes: vehicle identity: 15 categories, each having 25-160 instances; pose; and backgrounds: over 14 for each identity: projecting vehicles in relevant contexts. Further details are in the Appendix. We partition the 100-d latent space among attributes as: 60 for identity, 20 for pose, and 20 for background. iLab-20M has limited attribute combinations (identity shows only in relevant background; e.g., cars on roads but not in deserts), GZS-Net can disentangle these three attributes and reconstruct novel combinations (e.g., cars on desert backgrounds) Fig. 5 shows qualitative generation results.

We compare against (AE+DS), confirming that maintains discriminative information, and against two state-of-the-art GAN baselines: starGAN (Choi et al., 2018) and ELEGANT (Xiao et al., 2018). GAN baselines are strong in knowing *what to change* but not necessarily *how to change it*: Where change is required, pixels are locally perturbed (within a patch) but the perturbations often lack global correctness (on the image). See Appendix for further details and experiments on these GAN methods.

**RaFD** (Radboud Faces Database, Langner et al., 2010): contains pictures of 67 models displaying 8 emotional expressions taken by 5 different camera angles simultaneously. There are 3 attributes: identity, camera position (pose), and expression. We partition the 100-d latent space among the attributes as 60 for identity, 20 for pose, and 20 for expression. We use a 80:20 split for train:test, and use GZS-Net to synthesize images with novel combination of attributes (Fig. 6). The synthesized images can capture the corresponding attributes well, especially for pose and identity.

## 6 QUANTITATIVE EXPERIMENTS

### 6.1 QUANTIFYING DISENTANGLEMENT THROUGH ATTRIBUTE CO-PREDICTION

Can latent features of one attribute predict the attribute value? Can it also predict values for other attributes? Under *perfect* disentanglement, we should answer *always* for the first and *never* for the second. Our network did **not** receive attribute information through supervision, but rather, through swapping. We quantitatively assess disentanglement by calculating a model-based confusion matrix between attributes: We analyze models trained on the Fonts dataset. We take the Test examples from Font, and split them 80:20 for $\text{train}_{DR}$:$\text{test}_{DR}$. For each attribute pair $j, r \in [1..m] \times [1..m]$, we train a classifier (3 layer MLP) from $g_j$ of $\text{train}_{DR}$ to the attribute values of $r$, then obtain the accuracy of each attribute by testing with $g_j$ of $\text{test}_{DR}$. Table 1 compares how well features of each attribute (row) can predict an attribute value (column): perfect should be as close as possible to Identity matrix, with off-diagonal entries close to random (i.e., $1 / |\mathcal{A}_r|$). GZS-Net outperforms other methods, except for (AE + DS) as its latent space was Directly Supervised for this particular task, though it shows inferior synthesis performance.

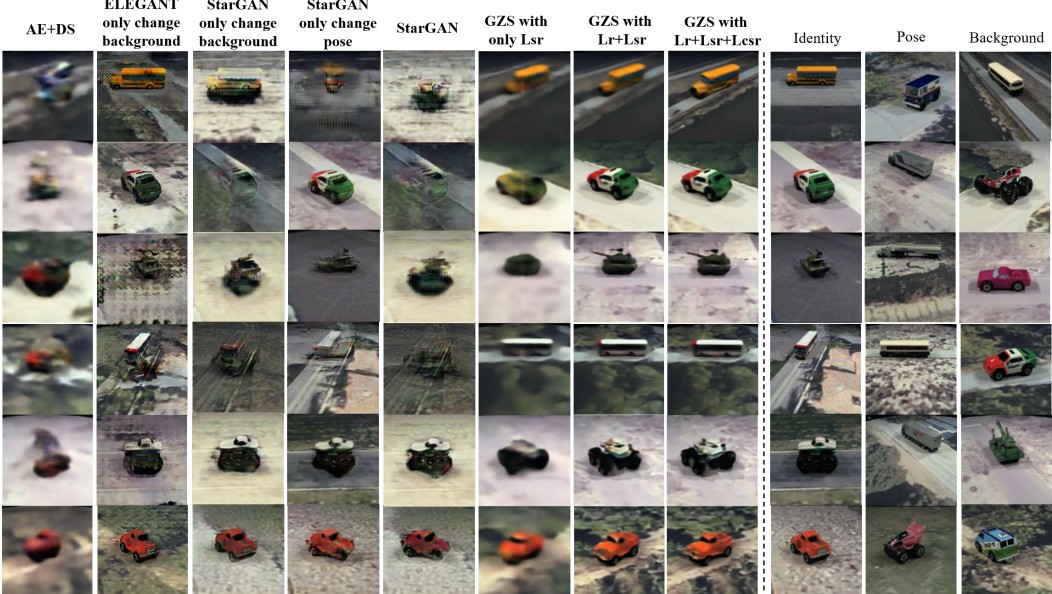

Figure 5: Zero-shot synthesis qualitative performance on ilab-20M. Columns left of the dashed line are output by methods: the first five are baselines, followed by three GZS networks. The baselines are: (1) is an auto-encoder with direct supervision (AE+DS); (2, 3, 4) are three GAN baselines changing only one attribute; (5) is starGAN changing two attributes. Then, first two columns by GZS-Net are ablation experiments: trained with part of the objective function, and the third column is output by a GZS-Net trained with all terms of the objective. starGAN of (Choi et al., 2018) receives one input image and *edit information* (explanation in Appendix Section B.4). ELEGANT uses identity and background images (in Appendix Section B.3, uses all input images). Others use all three inputs.

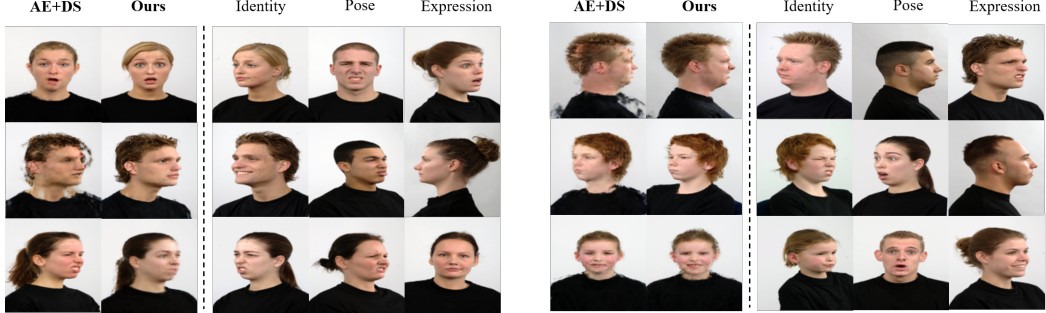

Figure 6: GZS-Net zero-shot synthesis performance on RaFD. 1-2 and 6-7 columns are the synthesized novel images using auto-encoder + Directly Supervision (AE+DS) and GZS-Net respectively. Remaining columns are training set images with their attributes provide.

## 6.2 DISTANCE OF SYNTHESIZED IMAGE TO GROUND TRUTH

The construction of the *Fonts* dataset allows programmatic calculating ground-truth images corresponding to synthesized images (recall, Fig. 4). We measure how well do our generated images compare to the ground-truth test images. Table 2 shows image similarity metrics, averaged over the test set, comparing our method against baselines. Our method significantly outperforms baselines.

## 6.3 GZS-NET BOOST OBJECT RECOGNITION

We show-case that our zero-shot synthesised images by GZS-Net can augment and boost training of a visual object recognition classifier (Ge et al., 2020). Two different training datasets (Fig. 7a) are tailored from iLab-20M, pose and background unbalanced datasets ($\mathcal{D}^{UB}$) (half classes with 6 poses per object instance, other half with only 2 poses; as we cut poses, some backgrounds are also

Table 1: Disentangled representation analysis. Diagonals are bolded.

| | GZS-Net | | | | | Auto-encoder | | | | | AE + DS | | | | | $\beta$-VAE + ES | | | | | $\beta$-TCVAE + ES | | | | |
|---|---|---|---|---|---|---|---|---|---|---|---|---|---|---|---|---|---|---|---|---|---|---|---|---|---|
| $\mathcal{A}$ ($|\mathcal{A}|$) | **C** | **S** | **FC** | **BC** | **St** | **C** | **S** | **FC** | **BC** | **St** | **C** | **S** | **FC** | **BC** | **St** | **C** | **S** | **FC** | **BC** | **St** | **C** | **S** | **FC** | **BC** | **St** |
| **C**ontent (52) | **.99** | .92 | .11 | .13 | .30 | **.48** | .60 | .71 | .92 | .06 | **.99** | .72 | .22 | .20 | .25 | **.02** | .35 | .11 | .19 | .01 | **.1** | .39 | .13 | .11 | .01 |
| **S**ize (3) | .78 | **1.0** | .11 | .15 | .36 | .45 | **.61** | .77 | .96 | .07 | .54 | **1.0** | .19 | .23 | .25 | .02 | **.38** | .29 | .11 | .01 | .02 | **.47** | .18 | .19 | .01 |
| **F**ont**C**olor (10) | .70 | .88 | **1.0** | .16 | .23 | .48 | .60 | **.67** | .95 | .06 | .19 | .64 | **1.0** | .66 | .20 | .02 | .33 | **.42** | .11 | .01 | .02 | .35 | **.21** | .13 | .01 |
| **B**ack**C**olor (10) | .53 | .78 | .21 | **1.0** | .15 | .53 | .63 | .64 | **.93** | .08 | .32 | .65 | .29 | **1.0** | .25 | .02 | .34 | .11 | **.86** | .01 | .03 | .40 | .24 | **.75** | .01 |
| **St**yle (100) | .70 | .93 | .12 | .12 | **.63** | .49 | .60 | .70 | .94 | **.06** | .38 | .29 | .20 | .20 | **.65** | .02 | .33 | .10 | .11 | **.02** | .02 | .33 | .10 | .08 | **.01** |

Table 2: Average metrics between ground-truth test image and image synthesized by models, conducted over the *Fonts* dataset. We report MSE (smaller is better) and PSNR (larger is better).

| | GZS-Net | AE+DS | $\beta$-TCVAE +ES | $\beta$-vae + ES | AE +ES |
|---|---|---|---|---|---|
| Average Mean Squared Error (MSE) | **0.0014** | 0.0254 | 0.2366 | 0.1719 | 0.1877 |
| Average Peak Signal-to-Noise Ratio (PSNR) | **29.45** | 16.44 | 6.70 | 9.08 | 7.9441 |

eliminated), as well as pose and background balanced dataset ($\mathcal{D}^{B}$) (all classes with all 6 poses per object instance).

We use GZS-Net to synthesize the missing images of $\mathcal{D}^{UB}$ and synthesize a new (augmented) balanced dataset $\mathcal{D}^{B\text{-}s}$. We alternatively use common data augmentation methods (random crop, horizontal flip, scale resize, etc) to augment the $\mathcal{D}^{UB}$ dataset to the same number of images as $\mathcal{D}^{B\text{-}s}$, called $\mathcal{D}^{UB\text{-}a}$. We show object recognition performance on the test set using these four datasets respectively. Comparing $\mathcal{D}^{B\text{-}s}$ with $\mathcal{D}^{UB}$ shows $\approx 7\%$ points improvements on classification performance, due to augmentation with synthesized images for missing poses in the training set, reaching the level of when all real poses are available ($\mathcal{D}^{B}$). Our synthesized poses outperform traditional data augmentation ($\mathcal{D}^{UB\text{-}a}$)

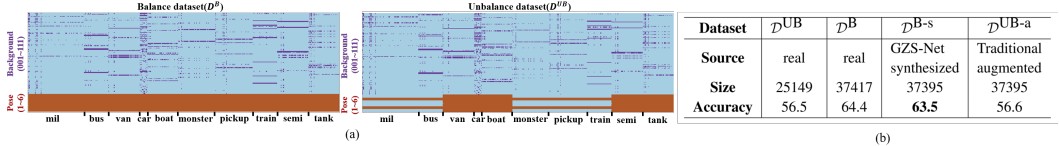

Figure 7: (a) Dataset details for training object recognition task, where the x-axis represents different identities (1004) and the y-axis represents the backgrounds (111) and poses (6) each purple and brown pixel means our dataset covers the specific combination of attributes. (b) object recognition accuracy (%) on 37469 test examples, after training on (augmented) datasets.

# 7 CONCLUSION

We propose a new learning framework, Group Supervised Learning (GSL), which admits datasets of examples and their semantic relationships. It provides a learner groups of semantically-related samples, which we show is powerful for zero-shot synthesis. In particular, our Group-supervised Zero-Shot synthesis network (GZS-Net) is capable of training on groups of examples, and can learn disentangled representations by explicitly swapping latent features across training examples, along edges suggested by GSL. We show that, to synthesize samples given a query with custom attributes, it is sufficient to find one example per requested attribute and to combine them in the latent space. We hope that researchers find our learning framework useful and extend it for their applications.

ACKNOWLEDGMENTS

This work was supported by C-BRIC (one of six centers in JUMP, a Semiconductor Research Corporation (SRC) program sponsored by DARPA), the Army Research Office (W911NF2020053), and the Intel and CISCO Corporations. The authors affirm that the views expressed herein are solely their own, and do not represent the views of the United States government or any agency thereof.

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

## APPENDIX

## A FONTS DATASET

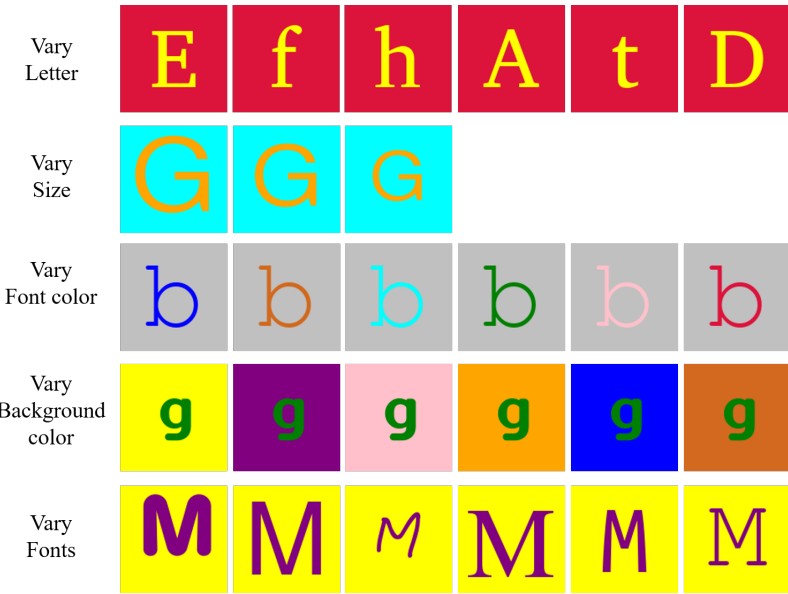

Figure 8: Samples from the Fonts dataset, a new parametric dataset we created by rendering characters under 5 distinct attributes. In each row, we keep all attributes the same but vary one.

Fonts is a computer-generated RGB image datasets. Each image, with $128 \times 128$ pixels, contains an alphabet letter rendered using 5 independent generating attributes: letter identity, size, font color, background color and font. Fig.1 shows some samples: in each row, we keep all attributes values the same but vary one attribute value. Attribute details are shown in Table 1. The dataset contains all

possible combinations of these attributes, totaling to 1560000 images. Generating attributes for all images are contained within the dataset. Our primary motive for creating the Fonts dataset, is that it allows fast testing and idea iteration, on disentangled representation learning and zero-shot synthesis.

You can download the dataset and its generating code from: `http://ilab.usc.edu/datasets/fonts` , which we plan to keep up-to-date with contributions from ourselves and the community.

Table 3: Attributes generating the Fonts dataset

| Attribute | Number of Attribute Values | Attribute Value Details |
|---|---|---|
| Letter | 52 | Uppercase Letters ($A$-$Z$) 
 Lowercase Letters ($a$-$z$) |
| Size | 3 | Small, Medium, Large 
 (80, 100, 120 pixel height respectively) |
| Font color | 10 | Red, Orange, Yellow, Green, Cyan 
 Blue, Purple, Pink, Chocolate, Silver |
| Background color | 10 | Red, Orange, Yellow, Green, Cyan 
 Blue, Purple, Pink, Chocolate, Silver |
| Font | 100 | Ubuntu system fonts 
 e.g. aakar, chilanka, sarai, etc. |

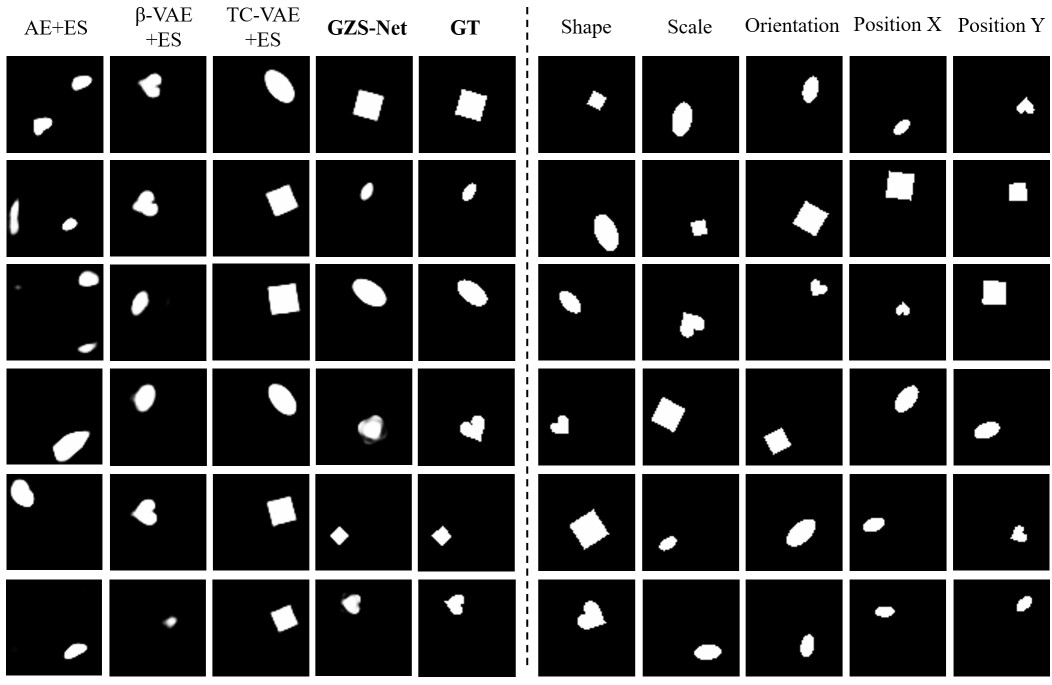

Figure 9: Zero-shot synthesis performance on dSprites. Columns 6-10 are input group images: from each, we want to extract one attribute (title of column). The goal is to combine the attributes to synthesize an new images. Columns 1-4 are synthesized images, respectively using: auto-encoder + Exhaustive Search (AE+ES), $\beta$-VAE + Exhaustive Search ($\beta$-VAE+ES), TC-VAE + Exhaustive Search (TC-VAE+ES) and GZS-Net respectively. The 5th column are ground truth (GT), which none of the methods saw during training or synthesis

## B BASELINES

### B.1 EXHAUSTIVE SEARCH (ES) AFTER TRAINING AUTO-ENCODER BASED METHODS

After training the baselines: standard Autoencoder, a $\beta$-VAE (Higgins et al., 2017), and TC-VAE (Chen et al., 2018). We want to search for the assignment between latent variables and attributes, as these VAEs do not make explicit the assignment. This knowing the assignment should hypothetically allow us to trade attributes between two images by swapping feature values belonging to the attribute we desire to swap.

To discover the assignment from latent dimension to attribute, we map all $n$ training images through the encoder, giving a 100D vector per training sample $\in \mathbb{R}^{n \times 100}$. We make an 80:20 split on the vectors, obtaining $X_{\text{train}_{ES}} \in \mathbb{R}^{0.8n \times 100}$ and $X_{\text{test}_{ES}} \in \mathbb{R}^{0.2n \times 100}$. Then, we randomly sample $K$ different partitionings $P$ of the 100D space evenly among the 5 attributes. For each partitioning $p \in P$, we create 5 classification tasks, one task per attribute, according to $p$: $\left\{ \left( X_{\text{train}_{ES}}[:, p_j] \in \mathbb{R}^{0.8n \times 20}, X_{\text{test}_{ES}}[:, p_j] \in \mathbb{R}^{0.2n \times 20} \right) \right\}_{j=1}^{5}$. For each task $j$, we train a 3-layer MLP to map $X_{\text{train}_{ES}}[:, p_j]$ to their known attribute values and measure its performance on $X_{\text{test}_{ES}}[:, p_j]$. Finally, we commit to the partitioning $p \in P$ with highest average performance on the 5 attribute tasks. This $p$ represents our best effort to determine which latent feature dimensions correspond to which attributes. For zero-shot synthesis with baselines, we swap latent dimensions indicated by partitioning $p$. We denote three baselines with this **E**xhaustive **S**earch, using suffix **+ES** (Fig. 4).

### B.2 DIRECT SUPERVISION (DS) ON AUTO-ENCODER LATENT SPACE

The last baseline (AE+DS) directly uses attribute labels to supervise the latent disentangled representation of the auto-encoder by adding auxiliary classification modules. Specifically, the encoder maps an image sample $x^{(i)}$ to a 100-d latent vector $z^{(i)} = E(x^{(i)})$, equally divided into 5 partitions corresponding to 5 attributes: $z^{(i)} = [g_1^{(i)}, g_2^{(i)}, \ldots, g_5^{(i)}]$. Each attribute partition has a attribute label, $[y_1^{(i)}, y_2^{(i)}, \ldots, y_5^{(i)}]$, which represent the attribute value (e.g. for font color attribute, the label represent different colors: red, green, blue,.etc). We use 5 auxiliary classification modules to predict the corresponding class label given each latent attribute partitions as input. We use Cross Entropy loss as the classification loss and the training goal is to minimize both the reconstruction loss and classification loss.

After training, we have assignment between latent variables and attributes, so we can achieve attribute swapping and controlled synthesis (Fig. 4 (AE+DS)). The inferior synthesis performance demonstrates that: The supervision (classification task) preserves discriminative information that is insufficient for photo-realistic generation. While our GZS-Net uses one attribute swap and cross swap which enforce disentangled information to be sufficient for photo-realistic synthesis.

### B.3 ELEGANT (XIAO ET AL., 2018)

We utilize the author's open-sourced code: `https://github.com/Prinsphield/ELEGANT`. For ELEGANT and starGAN (Section B.4), we want to synthesis a target image has same identity as *id provider image*, same background as *background provider image*, and same pose as *pose provider image*. To achieve this, we want to change the background and pose attribute of id image.

Although ELEGANT is strong in making image transformations that are local to relatively-small neighborhoods, however, it does not work well for our datasets, where image-wide transformations are required for meaningful synthesis. This can be confirmed by their model design: their final output is a pixel-wise addition of a residual map, plus the input image. Further, ELEGANT treats all attribute values as binary: **they represent each attribute value in a different** part of the latent space, whereas **our method devotes part of the latent space to represents all values for an attribute**. For investigation, we train dozens of ELEGANT models with different hyperparameters, detailed as:

- For iLab-20M, the pose and background contain a total of 117 attribute values (6 for pose, 111 for background). As such, we tried training it on all attribute values (dividing their latent space among 117 attribute values). We note that this training regime was too slow and the loss values do not seem to change much during training, even with various learning rate choices (listed below).

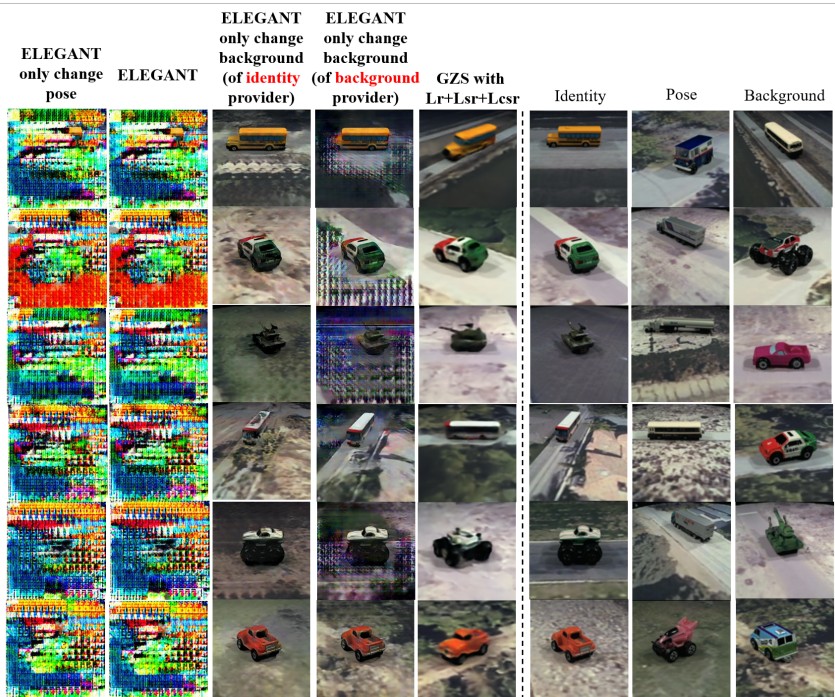

Figure 10: Zero-shot synthesis for ELEGANT, investigating the modification of pose and background attributes on an identity image. Details are in Section B.3.

- To reduce the difficulty of the task for ELEGANT, we ran further experiments restricting attribute variation to **only 17 attribute values** (6 for pose, 11 for background) and this shows more qualitative promise than 117 attributes. This is what we report.

- Fig 10 shows that ELEGANT finds more challenge in changing the pose than in changing the background. We now explain how we generated Columns 3 and 4 of Fig 10 for modifying the background. We modify the latent features for the identity image before decoding. Since the *Identity input image* and the *Background input image* have known but different background values, their background latent features are represented in two different latent spaces. One can swap on one or on both of these latent spaces. Column 3 and 4 of Fig.10 swap only on one latent space. However, in Fig. 5 of the main paper, we swap on both positions. We also show swapping only the pose attribute (across 2 latent spaces) in Column 1 of Fig.10 and swapping both pose and background in Column 2.

- To investigate if the model's performance is due to poor convergence of the generator, we qualitatively assess its performance on the **training set**. Fig. 11 shows output of ELEGANT on training samples. We see that the reconstruction (right) of input images (left) shows decent quality, suggesting that the generator network has converged to decently good parameters. Nonetheless, we see artefacts in its outputs when amending attributes, particularly located in pixel locations where a change is required. This shows that the model setup of ELEGANT is aware that these pixel values need to be updated, but the actual change is not coherent across the image.

- For the above, we applied a generous sweep of training hyperparameters, including:
  - **Learning rate**: author's original is 2e-4, we tried several values between 1e-5 and 1e-3, including different rates for generator and discriminator.
  - **Objective term coefficients**: There are multiple loss terms for the generator, adversarial loss and reconstruction loss. We used a grid search method by multiplying the original parameters by a number from [0.2, 0.5, 2, 5] for each of the loss terms and tried several combinations.
  - **The update frequency** of weights on generator (G) and discriminator (D). Since D is easier to learn, we performing $k$ update steps on G for every update step on D. We tried $k = 5, 10, 15, 20, 30, 40, 50$.

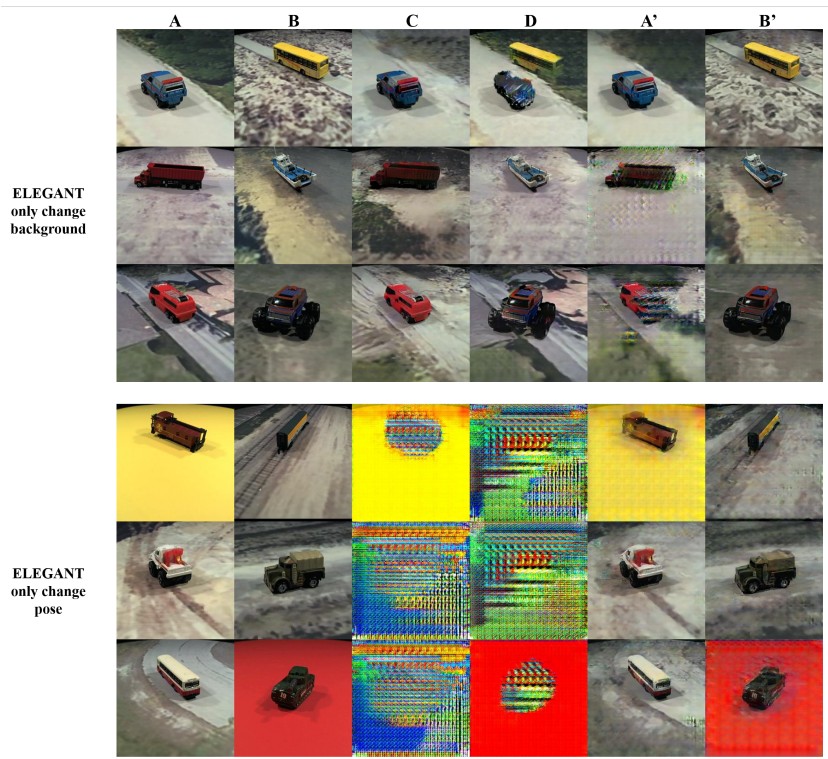

Figure 11: Training performance of ELEGANT. The left 2 columns (A and B) are input image. The following 4 columns are the generated synthesized images: A' and B' are reconstructions of the input (acceptable quality, suggesting convergence of generator), whereas C and D are the result of swapping features before the generator: C (/ D) uses the latent features of A (/ B) except by swapping background with B (/ A). All (C, D, A', B') share the same generator.

We report ELEGANT results showing best qualitative performance.

Overall, ELEGANT does not work well for holistic image manipulation (though works well for **local image edits**, per experiments by authors (Xiao et al., 2018)).

### B.4    STARGAN (CHOI ET AL., 2018)

We utilize the author's open-sourced code: `https://github.com/yunjey/stargan`. Unlike ELEGANT (Xiao et al., 2018) and our method, starGAN only accepts one input image and an edit information: the edit information, is **not extracted from another image** – this is following their method and published code.

## C    ZERO-SHOT SYNTHESIS PERFORMANCE ON DSPRITES DATASET

We qualitatively evaluate our method, Group-Supervised Zero-Shot Synthesis Network (GZS-Net), against three baseline methods, on zero-shot synthesis tasks on the dSprites dataset.

### C.1    DSPRITES

dSprites (Matthey et al., 2017) is a dataset of 2D shapes procedurally generated from 6 ground truth independent latent factors. These factors are color, shape, scale, rotation, x- and y-positions of a sprite. All possible combinations of these latents are present exactly once, generating 737280 total images. Latent factor values (Color: white; Shape: square, ellipse, heart; Scale: 6 values linearly

spaced in [0.5, 1]; Orientation: 40 values in [0, 2 pi]; Position X: 32 values in [0, 1]; Position Y: 32 values in [0, 1])

## C.2 EXPERIMENTS OF BASELINES AND GZS-NET

We train a 10-dimensional latent space and partition the it equally among the 5 attributes: 2 for shape, 2 for scale, 2 for orientation, 2 for position $X$, and 2 for position $Y$. We use a train:test split of 75:25.

We train 3 baselines: a standard Autoencoder, a $\beta$-VAE (Higgins et al., 2017), and TC-VAE (Chen et al., 2018). To recover the latent-to-attribute assignment for these baselines, we utilize the *Exhaustive Search* best-effort strategy, described in the main paper: the only difference is that we change the dimension of Z space from 100 to 10. Once assignments are known, we utilize these baseline VAEs by attribute swapping to do controlled synthesis. We denote these baselines using suffix **+ES**.

As is shown in Figure 2, GZS-Net can precisely synthesize zero-shot images with new combinations of attributes, producing images similar to the groud truth. The baselines $\beta$-VAE and TC-VAE produce realistic images of good visual quality, however, not satisfying the requested query: therefore, they cannot do controllable synthesis even when equipped with our best-effort Exhaustive Search to discover the disentanglement. Standard auto-encoders can not synthesis meaningful images when combining latents from different examples, giving images outside the distribution of training samples (e.g. showing multiple sprites per image).

