# OpenReview forum: "Zero-shot Synthesis with Group-Supervised Learning"
_ICLR.cc/2021/Conference — ICLR 2021 Poster_

### Official Review · AnonReviewer4 · 2020-10-27
**More analysis and experiments are needed**

**Rating:** 6
**Confidence:** 4

**Review:**

This paper proposes a Group-Supervised learning method for zero-shot synthesis. This method can generate multi-attributes images through the GZS-Net.
However, the authors may pay attention to the following questions:
1. The authors should check for grammar mistakes, e.g.,”Fonts is a...” ->”Fonts are...”.
2. From the description, the Group-Supervised learning method is just a special data sampling manner used in training. Besides, the total framework is commonly used in GANs for attribute control generation.
3. The contrast methods are too simple. The reviewer thinks the authors should compare their method with some new works, e.g., “ELEGANT: Exchanging Latent Encodings with GAN for Transferring Multiple Face”.
4. In Section 5.3, the reasons for the results in Table 1 are not analyzed. Why is the accuracy of contrast method AE-DS higher than GZS-Net?
5. The author should add more quantitative analysis. Such as comparing the FID/IS values of generated results using different methods.

---

> ### Author Response · Authors · 2020-11-14
> **RE: Disentanglement for synthesis VS discrimination; GZS-Net has partitioned latent space; GAN experiments**
>
>  1. **Grammar**: *Fonts* is a name of our dataset. We made it italic. We also corrected the other grammar mistakes that we observed.
>  2. **Our method is a Sampling method**:  You are right GSL samples examples in a special way (that is aware of, which examples share attribute values), but *additionally and crucially*, the model is aware of these attributes (the latent space is partitioned among attributes) and training algorithm is aware of the group nature (semantic relations) within the data, as it swap values in the latent space, as summarized by the training algorithm listed in the paper. Ours, has merits over the GAN-based approaches: Our method admits a simpler training pipeline: one learning rate, auto-encoder architecture with latent swap -- versus GAN, that require carefully choosing 2 learning rates: one for G one for D; gradient clipping; and overall provide no guarantees for convergence. Nonetheless, we show results for starGAN (see comment on top, addressing all reviewers).
>  3. We are now setting up to run ELEGANT, but we wanted to respond sooner (and we will respond again, once we have results on ELEGANT). However, we feel that ELEGANT will suffer from the same issue as starGAN, in that it will be good at local image transformations but not global ones (i.e. might also fail in rotating an object, while maintaining semantic correctness). This is supported by their architecture (see Figure 6 of ELEGANT): their attribute transfer takes the form of **element-wise addition** in the pixel space (which works well for local edits). Nonetheless, it is *possible* for someone to train a GAN-based approach using GSL, however, we leave this as future work. As an initial contribution, we want to show that even simple NN architectures, such as autoencoders, suffice for our tasks when using GSL.
>  4.  Table1 shows one way to evaluate disentanglement in the latent space. Good performance result in table1 shows that the information in the latent space is good enough information for discrimination across different attribute values: this does not imply good and realistic to image synthesis (shown in figure 4,5 and 6). To comment on this, we added the sentence in the text when we mention Table 1: _Directly supervising an autoencoder on attributes helps in classifying them, but shows inferior synthesis performance_.
> The reason why AE+DS performs well on table1, because this exact descrimination is used in its training objective.
>  5. We now have conducted quantitative experiments, please see the comment addressing all reviewers, on top.

---

> > ### Author Response · Authors · 2020-11-24
> > **ELEGANT conclusive results**
> >
> > We ran ELEGANT extensively. You may view results Figure 5 in the updated paper version with a detailed investigation of ELEGANT in Appendix.

---

> > > ### Comment · AnonReviewer4 · 2020-11-24
> > > **The paper could be accepted**
> > >
> > > Thanks for the response and the new experiments. The information has addressed most of my concerns, so I recommend to accept this paper.

---

### Official Review · AnonReviewer3 · 2020-10-27
**Recommendation to Accept**

**Rating:** 7
**Confidence:** 3

**Review:**

Summary:

The paper proposed a new training framework, namely GSL, for novel content synthesis. And GSL enables learning of disentangled representations of tangible attributes and achieve novel image synthesis by recombining those swappable components under a zero-shot setting. The framework leverages the underlying semantic links across samples which could be instantiated as a multigraph. Cycle-consistent reconstruction loss as well as reconstruction loss are computed on synthetic samples from swapped latent representations.

Reasons for score:

I vote for accepting. Overall, I think the paper is well written and the idea of GSL leveraging the underlying semantic relationships between samples to learn disentangled representations of tangible attributes is pretty novel and interesting. Experimental result on several real and synthetic datasets are also promising.

Concerns:

1) How are hyperparameters {d_j}^m_{j=1} selected? It would be great if the authors could elaborate more about this.

2) Concerning the cycle attribute swap, how to guarantee the intermediate generated image to be reasonable instead of a trivial solution. Seems like no losses (possibly like adversarial loss) is directly added on the intermediate result.

---

> ### Author Response · Authors · 2020-11-14
> **Conducted ablation on Cycle-swap and clarifications**
>
>
>  1. Per-attribute latent dimensions were selected by some preliminary experiments, but they were mostly guided by intuition: for instance, we feel that the identity attribute for faces/vehicles should carry the most information and therefore be devoted the highest capacity. Nonetheless, this is an interesting question, so we have started running some experiments to change the dimension which we will add to supplementary material.
>  2. We ran ablation experiments for removing the cycle swap loss term (please refer to the comment, addressing all reviewers, on top). Indeed, results are downgraded when the loss is removed. As we write in the text, the two swaps should guarantee that there is no information loss. Our intuition tells us that these swaps are not _trivial_: if they were trivial (e.g., swapping latent values carrying no information), then we should not be able to reconstruct the inputs, especially that **all** latent values are eventually swapped (for different example pairs) -- therefore, no attribute latents carry zero information.
> We did not add adversarial loss on the intermediate result, however, it seems like a good idea and might improve the performance, especially for a small dataset. Nonetheless our method is currently fully supervised: it reduces the choices of hyperparameters, uses feed-forward auto-encoders.

---

> > ### Comment · AnonReviewer3 · 2020-11-24
> > **Recommend to accept**
> >
> > I have read the author's response and I think overall they have addressed my concerns. Therefore I am changing my score to recommend acceptance.

---

### Official Review · AnonReviewer1 · 2020-10-27
**Review of "ZERO-SHOT SYNTHESIS WITH GROUP-SUPERVISED LEARNING"**

**Rating:** 7
**Confidence:** 4

**Review:**

I. Summary of paper.

The authors propose GZS-Net, a model based on an auto-encoder that synthesizes realistic images by exploring the semantic relationships between group images. The model is trained with a family of objective functions expressed on groups of examples.

It first maps training images into disentangled latent representations, which can be decomposed into multiple components, which in the specific implementation correspond to semantic attributes. Components are then recombined to form novel images that can correspond to previously unseen combinations of attributes.

The main experimental setup considered is zero-shot image synthesis: having seen different combinations of attributes, the model relies on disentangling attribute representations to synthesize novel combinations of previously seen attributes. Good performance is achieved on the Fonts, ilab-20M and RaFD datasets. In addition, the authors perform a more detailed analysis of their proposed approach and show that it results in good downstream performance on an object recognition task.

II. Strong and weak points.

Positive:
- The proposed task, zero-shot synthesis, is interesting.
- Overall, the paper is well-written and clear
- Good results against the selected baselines
- The novelty of the approach itself.

Negative:
- The following justifies some of the comments below. The authors claim GAN-related training is unstable with significant mode collapse. This is simply not the case currently, and in particular in the relatively small datasets considered. A comparison with e.g. cycle-GAN is fully warranted.
- The qualitative comparisons are limited. Yes, the model can generate good-looking images on what are mostly toy tasks. But these generations are commonly compared only to a relatively weak baseline (AE + DS) in fig 6, 5. Cycle GANs would work in this setting, and I supposed might be very competitive.
- The paper suffers from a general lack of (serious) quantitative comparisons. The authors provide generated images but the single results table on disentanglement does not contain models that would fare well in this setting.

III. Recommendation and justification.

I feel the paper is borderline, with a slight inclination to reject. On one hand, an interesting task is presented and the writing/clarity/experiments make this submission an interesting submission from a scientific standpoint. On the other hand, a lot of experimental work is needed to show this approach offers a benefit compared to serious baselines.

IV. Advice for author response.

I like this paper conceptually and would be prepared to raise my score. I'm interested in the following, if possible:
- Is there a way for the authors to provide an indication of sample quality versus adversarial baselines (cycleGAN, starGAN or some related method e.g.) to see if results hold?
- On the disentanglement side, is there a way to extend Table 1 to show whether the model performs well when compared to models that are good at disentangling, e.g. beta-VAE family.

If not feasible to include new experiments, please if possible at least discuss the issue wrt. baselines.

V. Improving the paper.

On the link between zero-shot learning, disentanglement, and generative models, I would suggest including [1, 2]. To be clear, these are just suggestions, choosing to include them or not will have no impact on my rating.

References

[1] Sylvain, Tristan, Linda Petrini, and Devon Hjelm. "Locality and Compositionality in Zero-Shot Learning." International Conference on Learning Representations. 2019.

[2] Higgins, Irina, et al. "SCAN: Learning Hierarchical Compositional Visual Concepts." International Conference on Learning Representations. 2018.

Edit: I have read the author's response. They have responded to the questions I had, and they address many of the concerns I had. I am now raising


Edit: updating score, and recommending acceptance as per my response to the author's rebuttal.

---

> ### Author Response · Authors · 2020-11-14
> **We conducted more experiments**
>
> ## Experiments
> We understand that your biggest concern, that you feel our work falls short, in terms of experimentation. Therefore, we ran more experiments: please view the comment addressing all reviewers.
>
>  * Specifically, rather than cycleGAN which work well for transferring *one attribute* value: in our setting, we would have to train a pair of generator and discriminator for each attribute value pairs (E.g., we have 111 background attribute values for vehicles, it would require us to train thousands of GAN models). Therefore,  we chose starGAN, its their authors show that it can be used to edit *multiple attributes* (i.e., more similar to ours).
>  * We also ran Quantitative results as you requested.
>  * We have extended table 1
> For all of these results, please refer to the comment above, addressing all reviewers.
>
> ## References
> Thank you for the references, we will be adding them to the manuscript.

---

> > ### Comment · AnonReviewer1 · 2020-11-14
> > **Follow-up on author response**
> >
> > I have read the author's response and agree that overall they have addressed the concerns I had. As in particular the lack of important baselines/experiments was a major point for me and it is now addressed, I am changing my score and recommend acceptance.
> >
> > Minor points: should the paper be accepted, the authors will have more space to add the results they are working on (wrt. the comment on lack of space in the general point addressed to all reviewers). As a suggestion, I would recommend the authors generate more samples (not now but after the rebuttal, I understand it takes a lot of time to train these baselines fully) to include in the appendix, as a means for a reader to get a better understanding of how different models perform.

---

### Official Review · AnonReviewer2 · 2020-10-28
**A nice contribution on disentangled representation learning and novel view synthesis**

**Rating:** 8
**Confidence:** 3

**Review:**

Summary:
This paper proposes Group-Supervised Learning (GSL) that can learn disentangled representations by swapping components in latent space, and enable one-shot novel view synthesis.  They created large dataset to evaluate their method, and demonstrate its effectiveness in controllable image synthesis, and disentanglement, outperforming existing baselines.

Pros:
1. This paper is good-written and clear to follow. The authors can demonstrate their idea well in Section 3 and 4.
2. Swapping attributes is an interesting and novel idea to encourage disentangled representation learning, which are easy to implement and could have wide applications in many fields.
3. The experiments are extensive, and the results are able to support their claims.

Cons:
1. In Equation (4), can you explain the defination of  separate losses, L_r, L_sr and L_csr along with the equation? They are not mentioned above, but simply in the algorithm 1.

2. If the cycle loss is removed, will the performance degrade dramatically? Can you please offer ablations for each loss?

---

> ### Author Response · Authors · 2020-11-14
> **Clarified loss terms and conducted Ablations**
>
> ## Cons
>  *  We now have clarified the meaning of those terms in the writeup, as some readers might prefer to not pay as much attention to the algorithm listing. After the equation defining $\mathcal{L}$, we added the text:
> > where $L_\textrm{r}$, $L_\textrm{sr}$ and $L_\textrm{csr}$, respectively are the reconstruction, swap-reconstruction, and cycle swap reconstruction losses.
>
>  *  We are now conducting ablation experiments, thank you for the suggestion! We have results for removing the cycle loss (the term $L_\textrm{csr}$). Removing the term, worsens quality (photos look a little more blurry) and achieves worse quantitative metrics -- in addition to the tables above, we have created this quantitative comparison on this ablation:
>
> |  ilab-20M  | GZS-Net (without cycle loss)  $\ \ $ | GZS-Net (with cycle loss)  $\ \ $ |
> |---:    |:---:|:---:|
> |  Average MSE  $\ \ $  | 0.0073 |0.0046 |
> |  Average PSNR  $\ \ $  | 21.55 | 23.55 |
>
> Training with only swap reconstruction loss shows much worse qualitative results, and we will update the table (and finally, merge all tables into the paper, either in the appendix or in the main text, if we get a 9th page allowance with your acceptance recommendation). Note that removing _both_ swap reconstruction and cycle-swap reconstruction, would cast our method becomes to an autoencoder, and we have experiments on autoencoder in the paper.

---

> > ### Comment · AnonReviewer2 · 2020-11-24
> > **The authors' response**
> >
> > I have read the authors' response. They have addressed my concerns. I would like to recommend this paper to be accepted.

---

### Author Response · Authors · 2020-11-14
**Conducted more experiments**

We would like to thank our reviewers, which put considerable time and thoughts for helping improve our paper. Thank you, for giving back to the scientific community, especially during difficult times worldwide. This response is meant for all reviewers, we are additionally responding to each reviewer individually.

It has taken us a long time to respond as we were working hard on making new experiments, based on common concerns from reviewers and we address that here. We ran a SOTA GAN baseline and two ablation experiments:
1. We ran starGAN, as it is a state-of-the-art approach for amending visual attributes onto images, e.g., changing hair color or facial expression. On one hand, we find starGAN really strong in carrying local image transformations (e.g. changing skin tone or hair texture). On the other hand, our method better maintains global information: when rotating the main object, the scene also rotates with it, in a semantically correct way.
1. We run two ablation experiments: removing the cycle loss term, from the objective function. We see that having it is necessary. In addition, we also have an ablation only keeping the swap reconstruction loss. We get best results if we keep all three terms in the objective.

## Details on Experiment Results
 * **starGAN**: We conducted qualitative and quantitative results using using the authors’ published code of starGAN (). We can add the results below to appendix, if necessary, or use them to replace Figure 5 and update Table 1, in the main text, with your recommendation. As-is, merging the results below into the paper **would us over the 8-page limit allowed for the review**. Nonetheless, we host updated Fig5 on anonymous cloud share:  https://imgur.com/a/R3eI8ta . Note: starGAN has been training for 2 days, finishing >300K steps (size 32), and it is still training -- it seems converged, but we will update the figures especially if we notice any improvements.
 * **Ablation**: We ran two ablation experiments, which are reflected in the updated Fig5 anonymous link above. (i) Removing the cycle loss term, which is an unsupervised term that artificially operates on more data, as it operates on all example pairs, regardless of their attribute values. (ii) in another ablation, we keep only the swap reconstruction loss. We see that all loss terms are useful to give us better qualitative and quantitative performance
 * **Quantitative comparisons**: We conducted Quantitative experiments over the *Fonts* dataset. We decided to chose the Fonts dataset for 3 reasons: it is fast to train on, in addition, the ground-truth is always known (computable), and finally, other datasets there might be multiple-plausible ground truths (e.g. the texture does not have to be exact, for vehicle background or face skin tone). Results are:

|    | GZS-Net  $\ \ $ | AE+DS  $\ \ $ | $\beta$-TCVAE + ES  $\ \ $ |  $\beta$-vae + ES  $\ \ $ | AE +ES   $\ \ $ |
|---:    |:---:|---|---|---|---|
|  Average MSE | 0.0014  | 0.0254  | 0.2366  | 0.1719  | 0.1877  |
| Average PSNR  | 29.45  | 16.44  | 6.70  |  9.08 | 7.9441  |


Finally, we are extending Table 1 to also include $\beta$-VAE and  $\beta$-TCVAE:

|  $\beta$-VAE  | Content  $\ \ $ | Size  $\ \ $ | Font color  $\ \ $ |  Back color  $\ \ $ | Style  $\ \ $ |
|---:    |:---:|---|---|---|---|
|  Content  $\ \ $  |  **0.02** | 0.35 | 0.11 | 0.19 | 0.01 |
|  Size  $\ \ $  | 0.02 | **0.38** | 0.29 | 0.11 | 0.01 |
|  Font  $\ \ $  | 0.02 | 0.33 |  **0.42** | 0.11 | 0.01 |
|  Back  $\ \ $  | 0.02 | 0.34 | 0.11 |  **0.86** | 0.01 |
|  Style  $\ \ $  | 0.02 | 0.33 | 0.1 | 0.11 |  **0.02** |



|  $\beta$-TCVAE  | Content  $\ \ $ | Size  $\ \ $ | Font color  $\ \ $ |  Back color  $\ \ $ | Style  $\ \ $ |
|---:    |:---:|---|---|---|---|
|  Content  $\ \ $  | **0.1** |0.39 |0.13 |0.11 |0.01  |
|  Size  $\ \ $  | 0.02 | **0.47** | 0.18 |0.19 | 0.01 |
|  Font  $\ \ $  | 0.02 |0.35 | **0.21** |0.13 | 0.01 |
|  Back  $\ \ $  |0.03  |0.4 | 0.24 | **0.75** | 0.01 |
|  Style  $\ \ $  | 0.02  | 0.33| 0.1 | 0.08 | **0.01** |

---

### Author Response · Authors · 2020-11-24
**Updated paper: Experiments, Related work, Appendix.**

After extensive training of ELEGANT and starGAN (more experiments still ongoing), we are updating the rebuttal paper. Some of the results have already been shown in our previous rebuttal response (on Nov 13). We included all concluded experiments onto our paper (it is now at 9 pages, per rebuttal guidelines). We will pin-point the changes:

 1. (*small change*) Clarify the terms of objective function, Section 4.3
 1. (*medium change*) update the Related Work (Section 2), the Subsection "Conditional Synthesis", to discuss starGAN and ELEGANT as pointed by reviewers.
 1. (*large change*) Experiments.
    * Now we have two sections: Qualitative Experiments (Section 5) and Quantitative Experiments (Section 6).
    * Section 5 contains an updated Figure 5, including starGAN and ELEGANT baselines.
    * Section 6 contains additional Section 6.2  with image similarity metrics (of ground truth VS synthesized image, for Fonts dataset) with results in Table 2. It contains also updated Table 1 and accompanying explanation.
 1. (*large change*): Added 2 sections to appendix, explaining how we ran ELEGANT and starGAN and shows detailed results for ELEGANT.
 1. (*small changes*) throughout the paper, fixed small typos, grammar mistakes, writing clarities, that we located.

---

### Decision · Program_Chairs · 2021-01-07
**Final Decision**

**Decision:**

Accept (Poster)

**Comment:**

This paper presents a zero-shot generation approach by disentangling representations into swappable components (each component corresponding to an attribute) and then conditioning on any desired combination of attributes to do zero-shot synthesis of samples containing those attributes.

There were some concerns raised in the original reviews which the authors have addressed in the rebuttal and the revised submission. Post the discussion phase, all reviewers see merit in the proposed ideas and unanimously recommend acceptance. Based on my own reading of the paper and the reviews/author responses, I agree with the assessment.